# Viral Enteritis in Solid-Organ Transplantation

**DOI:** 10.3390/v13102019

**Published:** 2021-10-07

**Authors:** Anum Abbas, Andrea J. Zimmer, Diana Florescu

**Affiliations:** Division of Infectious Diseases, University of Nebraska Medical Center, Omaha, NE 68198, USA; andreaj.zimmer@unmc.edu (A.J.Z.); dflorescu@unmc.edu (D.F.)

**Keywords:** viral enteritis, cytomegalovirus, norovirus, adenovirus, rotavirus, organ transplantation, diarrhea

## Abstract

Solid organ transplant recipients are at increased risk for infections due to chronic immunosuppression. Diarrhea is a commonly encountered problem post transplantation, with infectious causes of diarrhea being a frequent complication. Viral infections/enteritides in solid organ transplant recipients often result from frequently encountered pathogens in this population such as cytomegalovirus, adenovirus, and norovirus. However, several emerging viral pathogens are increasingly being recognized as more sensitive diagnostic techniques become available. Treatment is often limited to supportive care and reduction in immunosuppression, though antiviral therapies mayplay a role in the treatment in certain diseases. Viral enteritis is an important entity that contributes to morbidity and mortality in transplant recipients.

## 1. Introduction

Solid-organ transplant (SOT) recipients face unique infectious risks in the setting of chronic immunosuppression. Diarrhea is a commonly encountered problem in SOT recipients, with prevalence in the range of 20–50% [1]. Diarrhea often leads to dehydration and malnourishment but can potentially cause organ rejection and even death. Causes of diarrhea after transplantation encompass a wide array of both infectious (bacterial, parasitic, and viral pathogens) and noninfectious etiologies. Viral enteritis in SOT recipients accounts for a significant proportion of infectious enteritis, resulting in morbidity and mortality. Frequent pathogens include norovirus, adenovirus, rotavirus, and cytomegalovirus (CMV), but as more sensitive diagnostic techniques become available, several emerging viral causes of enteritis are being increasingly recognized. This review focuses on commonly encountered causes of viral enteritis, as well as emerging viral infections causing enteritis in SOT recipients.

## 2. Viral Enteritis

### 2.1. Cytomegalovirus

CMV, a double-stranded, enveloped DNA virus and member of the *Betaherpesvirinae* subfamily, is endemic worldwide, with approximately 50% seroprevalence among adults in the United States [2,3]. CMV is the most common opportunistic infection afflicting SOT recipients, with enterocolitis being the most frequent form of end-organ diseases [4,5,6]. Following primary infection, the virus establishes latency and reactivates post transplantation; it can also be transmitted with the allograft at the time of transplantation, resulting in donor-derived infection. The risk of CMV is highest when a seronegative (CMV IgG negative) individual receives an organ from an individual who is CMV seropositive (D+/R−) [4]. Rates of CMV infection and disease also vary according to allograft type, cumulative immunosuppression, and preventive strategies. Recipients of lung, composite tissue, and small bowel are at particularly high risk for CMV disease due to the high degree of immunosuppression and amount of lymphoid tissue transplanted with the allograft; CMV may manifest with an end-organ disease in the allograft [4]. In the absence of prophylaxis, CMV reactivations typically occur within the first 3–6 months post-transplantation [4]. CMV infection in SOT is associated with increased risk for other infectious complications, rejection, chronic allograft dysfunction, morbidity, and mortality [4,7].

The diagnosis of CMV gastrointestinal (GI) disease is based on the presence of upper and/or lower GI symptoms with CMV identified in biopsied tissue via histopathology, immunohistochemistry, or DNA hybridization from macroscopic mucosal lesions visualized during endoscopy [8]. Cases are classified as probable when upper and/or lower GI symptoms are present with CMV identified in tissue specimens, but macroscopic mucosal lesions are not seen [8]. The detection of CMV by PCR in blood or on tissue biopsy is not sufficient to diagnose tissue-invasive disease; the use of quantitative PCR on biopsies from the gut or stool has been considered [8,9,10].

Assessment of the level of immune response and risk for CMV infection in SOT recipients is gaining traction within clinical practice in the transplant community. CMV-specific cell-mediated immunity (CMV-CMI) may be utilized on a case-by-case basis to help guide management, including adjusting depth of immunosuppression, duration of CMV therapies, and prevention strategies [11,12,13,14]. Modalities available to assess CMV-CMI include enzyme-linked immunosorbent spot (ELISPOT), interferon-gamma release assays (IGRA), and intercellular cytokine staining (ICS) using flow cytometry [13,14,15,16,17].

Careful reduction in immunosuppression plays an important role in the control of CMV disease, particularly in cases of drug resistance or intolerance [4]. Parenteral ganciclovir (GCV) is the first-line treatment for CMV GI tissue-invasive disease, followed by a transition to oral valganciclovir upon improvement in symptoms and once a patient is reliably absorbing medications [4]. CMV enteritis should be treated for a minimum of 2 to 3 weeks, through the resolution of clinical symptoms, and until the virologic clearance is achieved [4,7]. Viral loads should be monitored while on therapy, generally weekly; if the viral load does not decline by 1 log_10_ after two weeks of treatment, genotypic resistance testing targeting *UL97* and *UL54* genes should be performed [18]. The *UL97* mutations confer different degrees of GCV resistance, but often, the anti-viral activity of foscarnet or cidofovir (CDV) is retained; both drugs have significant nephrotoxicity and electrolyte wasting associated with their use [4]. Combination therapy directed at multiple viral targets is not commonly used in clinical practice. The data regarding synergy for the combination of GCV-foscarnet against CMV is inconclusive, with very limited data regarding GCV resistant virus [19]. Mutations at *UL54* may result in variable resistance to GCV, foscarnet, and CDV, leading to limited therapeutic options [4].

Letermovir, an inhibitor of the subunit p*UL56* of the terminase enzyme complex, is approved for primary CMV prophylaxis in allogeneic stem cell transplant recipients [20]. Mixed results were reported with the use of letermovir for the treatment of CMV infections, and its use for the treatment of CMV infections is limited [20]. Maribavir, an oral antiviral agent that inhibits the *UL97* protein kinase, has the potential to redefine treatment for refractory and/or resistant CMV infections [21]. In a recently completed phase 3 study (SOLSTICE trial, NCT02931539), maribavir demonstrated superiority to the investigator-assigned therapy (GCV/valganciclovir, foscarnet, and CDV) in the treatment of refractory and/or resistant CMV infections (55.6% vs. 26.1%) [20]. Maribavir was associated with a lower incidence of treatment-related toxicities: lower rates of neutropenia vs. GCV/valganciclovir (1.7% vs. 25%) and acute kidney injury vs. foscarnet (1.7% vs. 19.1%) [20]. A recent study showed that maribavir is strongly antagonistic with GCV but had additive effects with foscarnet, CDV, and letermovir when tested against wild-type and mutant CMV viruses [22]. Non-pharmacologic therapy, including adoptive T-cell therapy, is being investigated as a treatment option, including in a recent prospective study, in which 13 SOT recipients were treated with autologous CMV-specific T cells, and 11 (84%) demonstrated some improvement in symptoms and viral levels [23].

### 2.2. Norovirus

Norovirus, a nonenveloped RNA virus from the *Caliciviridae* family, is prevalent worldwide and the most common cause of gastroenteritis globally. The high transmissibility and overall stability of the virus in harsh environments result in outbreaks, both in community and healthcare settings. While the incubation period is short, usually 24–48 h, the virus can be shed in the stool for several weeks or even months in immunocompromised hosts. Vomiting is usually the predominant symptom, though watery diarrhea is a common manifestation. Immunocompetent hosts typically have shorter and self-resolving episodes [24].

Immunocompromised hosts are at risk for developing recurrent or chronic infection with norovirus. A large retrospective study in kidney, heart, and lung transplant recipients showed that 22.8% of the patients infected with norovirus developed chronic infection; the median shedding period was 218 days (range 32–1164 days) [25]. Another study found that nausea at the time of presentation and CMV infection in the 90 days prior to norovirus diagnosis were independent risk factors for persistent diarrhea [26]. The authors hypothesize that nausea at the time of presentation may be a surrogate marker for high viral load. CMV infection itself might be a marker of immunosuppression and may reflect increased susceptibility to other infections, including norovirus. A single-center study among SOT patients identified norovirus as one of the most common infectious causes of diarrhea, 34.8%, which is a higher rate than previously reported among SOT recipients. The study retrospectively compared adult SOT patients with norovirus infection to patients with diarrhea due to other causes (most commonly identified causes were *C. difficile*, medications, and CMV) during the same period without norovirus infection; the study showed a significant protracted course of diarrhea in the norovirus group, with an average of 241 days vs. 75 days in the control group [27].

Unfortunately, there are no specific therapies approved for norovirus treatment at this time. Strategies focus on rehydration, antimotility agents, and reduction in immunosuppression, which may help decrease viral shedding [28]. Several treatments have been tried including oral and intravenous immunoglobulin, human breast milk, ribavirin, and nitazoxanide. Therapy with oral and intravenous immunoglobulin has generated mixed results with one study showing a decrease in stool output but no change in total time to resolution of diarrhea, length of stay, or cost [29].

Nitazoxanide, a thiazolide antimicrobial with antibacterial, antiviral, and antiparasitic properties, is thought to target cellular pathways involved in the synthesis of viral proteins [30]. A double-blind, placebo-controlled trial demonstrated treatment of norovirus infection with nitazoxanide significantly reduced the duration of illness; however, the study included only immunocompetent hosts in the outpatient settings, with mild-to-moderate illness [30]. Limited reports of immunocompromised patients successfully being treated with nitazoxanide exist, but further studies are needed to establish effectiveness and utility in this population [31]. A phase 2 multi-center, prospective double-blind study assessing nitazoxanide efficacy and safety in SOT and hematopoietic stem cell transplant recipients is currently underway (ClinicalTrials.gov Identifier: NCT03395405).

### 2.3. Rotavirus

Rotavirus, a nonenveloped double-stranded RNA virus in the *Reoviridae* family, is known to cause enteritis in infants and children under 5 years of age [32,33]. It has also been recognized as a problematic pathogen in SOT, particularly among intestinal, liver, and pediatric transplant recipients [18,32,34]. Rotaviruses are subdivided into seven serogroups (A–G), and Group A is most commonly identified in gastroenteritis among children and transplant recipients [32]. Clinical presentation in transplant recipients may include nausea, vomiting, abdominal pain, and watery diarrhea, which in severe cases, may precipitate renal failure [18]. A cohort study of intestinal transplants demonstrated a high rate (70%) of acute cellular rejection either concomitantly with rotavirus enteritis or within 90 days of infection [32,33]. Likely explanations for this correlation include allograft recognition prompted by activation of gut-associated lymphoid tissue or malabsorption of maintenance immunosuppressants related to vomiting and diarrhea [32,33]. However, elevated levels of tacrolimus have been reported in the setting of rotavirus, norovirus, and other infections causing enteritis, and are proposed to be a result of injured enterocytes’ reduced ability to perform pre-systemic CYP3A4 metabolism, thereby increasing systemic absorption [32,34,35].

Supportive therapy, including fluid resuscitation and electrolyte replacement, is the mainstay of treatment as currently there are no antiviral therapies available for rotavirus [5,18,32]. Two live attenuated vaccines are approved for use in the United States, but their use after transplantation is not recommended [5].

### 2.4. Adenovirus

Adenoviruses, nonenveloped lytic double-stranded DNA viruses from the *Adenoviridae* family, cause infections in both immunocompetent and immunocompromised hosts. Adenoviruses are classified into 7 subgroups (A–G), and these subgroups can be further divided into more than 50 serotypes. Serotypes have been identified to have tissue tropism; for example, gastroenteritis has been associated with serotypes 40, 41, and 52 [36]. Adenovirus is a common pediatric infection, after which it can establish latency in lymphoid tissue [37]. Infections do not have seasonal variability and the virus is transmitted via the fecal–oral route, droplets, and larger aerosols [38]. Definitions of adenoviral infection and disease have been proposed, though no formal consensus exists. Adenoviral infection is defined as the detection of the virus in stool, blood, urine, or respiratory tract without symptoms, and adenoviral disease is defined as the detection of adenovirus plus signs and symptoms that could be attributable to adenovirus in the absence of another diagnosis [38]. Disseminated disease is defined as the involvement of two or more organ systems and does not include viremia [38].

In SOT recipients, adenovirus causes a variety of clinical syndromes ranging from asymptomatic infection to life-threatening disseminated disease. Clinical manifestations vary with the type of allograft and commonly involve the graft [38]. Several factors have been associated with increased risk for the adenoviral disease after SOT including age, serologic mismatch, degree of immunosuppression, and allograft type [38]. The highest rates are reported in intestinal transplant recipients, possibly owing to significant lymphoid tissue transplanted with the allograft and higher degrees of immunosuppression.

The incidence of adenovirus infections in SOT recipients varies with organ type and differs among pediatric and adult SOT populations. Reported incidence among liver and kidney recipients ranges from 4.1 to 6.5%. In a study of viral enteritis in intestinal transplant recipients, adenoviral enteritis was noted in 25% of pediatric and adult intestinal transplant recipients with the median time from transplantation to diagnosis being 53 days [38].

As with the many other causes of viral enteritis, supportive care, including rehydration, antimotility agents, and reduction in immunosuppression, is the mainstay treatment [39]. Patients with adenoviral infection may clear it on their own, but those with invasive disease or disseminated infection will often require antiviral therapy. No antiviral agent has been studied in a randomized trial to establish efficacy in the SOT population, and no agents have been approved for the treatment of adenovirus infection.

CDV, a nucleotide analog of cytosine, has activity against all serotypes of adenovirus, and it is considered the standard of care for the treatment of adenovirus. However, its use is limited by severe nephrotoxicity and bone marrow suppression. CDV is usually considered for severe infections, with two different dosing options including the 1 mg/kg three times per week dose or 5 mg/kg per week dose, though a direct comparison of the two dosing strategies has never been carried out [38,40]. CDV is usually administered until testing from the site of positivity is negative, and complete resolution of the attributable symptoms is achieved.

An oral option and lipid conjugate of CDV, brincidofovir (Tembexa) was recently approved by FDA for the treatment of smallpox. As smallpox has been eradicated, brincidofovir was studied in patients that were infected with viruses closely related to the variola virus. It appears to have several advantages over CDV, including more potent antiviral activity and less nephrotoxicity. Most of the data regarding side effects, including severe diarrhea and liver toxicity (transaminitis), come from the adult hematopoietic stem cell transplant literature. Brincidofovir causes epithelial apoptosis and crypt injury that is very difficult to distinguish from graft-versus-host disease and mycophenolate mofetil toxicity, and this could inadvertently lead to increased immunosuppression. However, brincidofovir seems to be better tolerated in stem-cell and solid-organ transplant recipients [41]. In a recent report in pediatric SOT recipients (two kidneys, one kidney–liver, and one liver), treatment with brincidofovir was associated with mild GI symptoms, which resolved with completion of the therapy [42]. Brincidofovir was reported to be an effective salvage therapy in immunocompromised patients with adenovirus disease who failed CDV treatment. During the first week of therapy with brincidofovir, 61.5% of patients had a ≥1 log10 decrease in viral load; after 2 months, 69.2% demonstrated a virologic response [43]. An open-label study of IV brincidofovir in SOT recipients with adenovirus viremia was terminated due to a low enrollment rate (ClinicalTrials.gov Identifier: NCT03532035).

### 2.5. Epstein–Barr Virus

Epstein–Barr virus (EBV) is an enveloped, double-stranded DNA virus of the *Gammaherpesvirinae* subfamily. EBV is a commonly encountered virus and nearly all adults are seropositive by age 40. As with other herpesviruses, EBV establishes latency in its host and persists during the host’s lifetime, with its primary target being the naïve B cell [2]. Primary infection results in infectious mononucleosis, which is a self-limiting illness. In transplant recipients receiving potent T-cell immunosuppressing agents, EBV can cause post-transplant lymphoproliferative disorder (PTLD) [2]. There are a few reports of EBV enteritis in intestinal transplant recipients and has been observed more so in the pediatric population [44,45]. Recurrent enteritis in an auto hematopoietic stem cell transplant recipient has also been reported [46]. Currently, no guidance on management for these cases exists, and reduction in immunosuppression is the usual approach.

## 3. Emergent Viral Infections Causing Gastroenteritis

Some of the once unexplained gastroenteritides are now recognized to be caused by infectious agents (i.e., norovirus, sapovirus). The list of emerging enteric viral pathogens is constantly expanding, with the recent additions of astroviruses, bocavirus, cosavirus, torovirus, picobirnavirus, salivirus, bufavirus, Aichivirus, and Saffold virus. Since very few institutions perform electron microscopy, real-time PCR, or nested reverse transcription PCR for the diagnosis of gastroenteritis, these emerging viruses are probably underdiagnosed. As there is no specific antiviral therapy for any of these viruses, the treatment is based on rehydration, nutritional supplementation, and decreasing immunosuppression whenever possible.

### 3.1. Human Astrovirus

Human astroviruses (HastVs) are positive-sense, single-stranded RNA viruses belonging to the family *Astroviridae*. They are divided into classic human astroviruses (HAstV 1–8), MLB (HAstV-MLB 1–3), and VA (HAstV-VA 1–5) [47]. HAstV infections have been reported in immunocompromised patients, mainly in hematopoietic stem-cell transplant (HSCT) recipients. HAstVs can cause gastroenteritis, including mild, watery diarrhea with associated vomiting, fever, anorexia, and abdominal pain [48]. In immunocompromised patients, HastV-4, HastV-VA1/HMO-C/PS, and MBL2 astroviruses can cause disseminated disease, encephalitis, meningitis, and even death [49,50]. On pathology examination, mild changes in the jejunal and duodenal epithelial cells, villous blunting, and inflammation in lamina propria are observed [48]. HastVs can be detected by direct transmission electron microscopy, immune electron microscopy (IEM), or solid-phase immune electron microscopy (SPIEM), but these methods are laborious, time consuming, and not readily available in clinical practice [48]. Several commercial enzyme immunoassays (EIAs) can be used for HastV detection in stool specimens. Although the assays do not enable typing of the isolates, they are useful for rapid detection of the virus [48]. Reverse transcriptase PCR (RT-PCR) offers a low threshold of viral detection in stool specimens [48]. In certain cases, especially extraintestinal infections, a more definite diagnosis with deep sequencing often needs to be performed [50].

### 3.2. Human Bocaviruses

Human bocaviruses (HboVs) are nonenveloped, single-stranded DNA viruses belonging to the family *Parvoviridae.* HBoVs can cause respiratory infections and gastroenteritis in immunocompromised patients [51]. There are four HboV genotypes, of which genotypes 2–4 have been considered enteric pathogens [52,53]. The rate of HboV coinfections with other respiratory and gastrointestinal pathogens seems to be high, raising the question if HboV is a true pathogen or innocent bystander [52,53]. Recent data support that HboV, during its natural infection, can establish latency in host cells. However, the mechanisms of persistence, latency, and reactivation of HboV are not well understood [53]. In a recent study examining allogeneic HSCT recipients, HBoV was detected in both serum and fecal specimens, supporting its role as an enteric pathogen [52]. Most of the HBoV infections are asymptomatic, but in symptomatic patients, nausea, diarrhea, abdominal pain, and respiratory symptoms are common [51,54]. In two case reports, immunocompromised patients (a pediatric liver and stem cell transplant recipient) presented with severe diarrhea with profound hypovolemia and disseminated disease, a course later complicated by protracted shedding of the virus [54,55]. HBoV detection for prolonged periods in clinical specimens poses a concern for possible nosocomial dissemination in hospitalized patients [52]. Conventional PCR or nested RT-PCR can be used for diagnostic in various samples—stool, serum, urine, and bronchoalveolar lavage [56]. ELISA and EIA seem to be reliable when used to detect IgG and IgM antibodies and IgG affinity for diagnosis purposes [56].

### 3.3. Human Cosaviruses

Human cosaviruses (HCoSVs) are positive-sense, single-stranded RNA viruses belonging to the *Picornaviridae* family. They are classified into six species (A, B, C, D, E, F) [57]. The virus was initially detected in fecal samples from children with acute non-polio flaccid paralysis; subsequently, the virus was identified in stool from healthy children and adults. At present, the clinical importance of HCoSV remains unclear. Reports of HCoSV infections in solid organ transplantations are limited to 2 cases—one pediatric liver and one adult lung transplant recipient; these patients presented with fever, nausea, vomiting, persistent diarrhea, and abdominal pain [57]. The virus can be detected by RT-PCR or multiplex RT-PCR (along with rotavirus A, B, and C, adenovirus, astrovirus, norovirus GI and GII, sapovirus, Aichi virus, parechovirus, enterovirus, cosavirus, bocavirus, and Saffold virus) [58].

### 3.4. Sapovirus

Sapovirus, a nonenveloped, single-stranded RNA virus, belongs to the *Caliciviridae* family. Sapovirus has been identified as an important cause of gastroenteritis in immunocompromised patients after the introduction of reverse transcription-PCR (RT-PCR) [39,59]. Patients often present with vomiting, non-bloody watery diarrhea, and abdominal cramping; symptoms can last for several months, leading to severe dehydration and acute renal failure [59,60]. RT-PCR is the standard diagnostic method from stool samples, due to its high specificity and sensitivity and may be used for genotyping [61].

### 3.5. Torovirus

Torovirus, an enveloped, positive-stranded RNA virus, belongs to the *Tobaniviridae* family. Torovirus, an established agent of gastroenteritis in animals, is now recognized as the cause of gastroenteritis in pediatric patients [62]. In a prospective surveillance study in a tertiary pediatric center in Canada, Torovirus was the most common pathogen in nosocomial viral enteritis, occurring in all age groups and without any seasonal predilection [63]. A high proportion of these patients were immunocompromised (41%), on enteral feeding (59%), or had surgical or other invasive procedures (56%) [63]. A high proportion of patients present with fever and vomiting; bloody diarrhea is considered a hallmark feature of torovirus infection [63], and corona-like viral particles can be seen in the stool. The diagnosis is difficult to make since most centers do not have electron microscopy in correlation with enzyme immunoassay with Breda virus antiserum readily available [62].

### 3.6. Picobirnaviruses

Picobirnaviruses are small, nonenveloped, bi-segmented double-stranded RNA viruses belonging to the *Picobirnaviridae* family. These viruses can cause diarrhea as a single pathogen or co-infection with rotavirus, astrovirus, and *E. coli* [64]. Picobirnaviruses were detected in up to 40% of stool samples by RT-PCR assays and were commonly isolated from immunocompromised patients (mainly HIV-infected patients) [64]. In a small study of kidney transplant recipients, detection of picobirnaviruses was associated with high cyclosporine levels and prolonged immunosuppressive treatment [65].

### 3.7. Severe Acute Respiratory Syndrome Coronavirus 2

Severe acute respiratory syndrome coronavirus 2 (SARS-CoV-2), a positive-sense, single-stranded RNA virus, is a member of the *Coronaviridae* family. SARS-CoV-2 may play a direct role in the gastrointestinal symptoms in COVID-19 illness. Currently, the data regarding enterocolitis due to SARS-CoV-2 are limited. In a recently published study, diarrhea was more frequent in SOT recipients, compared with nontransplant patients (54% vs. 17%) [66]. In a case report, it was shown that SARS-CoV-2 can infect gastrointestinal epithelial cells expressing angiotensin-converting enzyme 2 receptors. Histopathology examination demonstrated excessive inflammatory response, intestinal mucosal erosions, and focal inflammatory necrosis with hemorrhage. Viral RNA was found by in situ hybridization in the intestinal tissue [67].

## 4. Conclusions

Viral enteritis continues to be a frequent and vexing complication following transplantation. In addition to their potentially prolonged and/or severe GI illnesses, the viral gastroenteritides threaten the already tenuous balance of immunosuppression, risking medication-induced toxicity and/or rejection with allograft dysfunction. Recent advances in pharmacologic and other treatment modalities are promising, but ongoing efforts to develop novel antiviral agents and prevention strategies are essential. Given the ever-changing nature of transplant medicine and the field of infectious diseases, clinicians and microbiologists working with SOT populations must remain constantly vigilant and investigative to optimize detection, prevention, and treatment of viral enteritis in transplant recipients.

## Data Availability

Not applicable.

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
