# Peer review of "Viral Enteritis in Solid-Organ Transplantation"

_viruses, 2021, doi:10.3390/v13102019_

Round 1

Reviewer 1 Report

Abbas et al. revised the occurrence of viral enteritis in solid organ transplant (SOT) recipients. The manuscript is interesting and generally well written.

I would suggest a revision of some sentences [for example, line 176 “Clinical manifestations vary with the type of allograft and do commonly involving the graft”; an incomplete sentence in line 187 “As with the many other causes of viral enteritis, supportive care including rehydration, antimotility agents and reduction in immunosuppression”;  line 191 “…no agents have been approved by the for (sic) treatment of adenovirus infection.”; line 205 “…regarding severe diarrhea and liver toxicity (transaminitis) comes [FROM?] adult hematopoietic stem cell transplant literature.”); redundancy in line 211 “…treatment with Brincidofovir was associated with mild gastrointestinal symptoms brincidofovir which...”; redundancy again in line 214 “During the first week of therapy with Brincidofovir, 61.5% of patients had a ≥ 1 log10 decrease in viral load after the first week of therapy” and line 237 “...and inflammation in lamina propria inflammatory are seen”].

Also, revise some typos (see for example, line 186 “trasnplant”, line 204 has two dots “..”).

Considering the recent approval of Brincidofovir (Tembexa) by FDA for the treatment of smallpox, I would suggest to include the information that since smallpox is eradicated, the effectiveness of Tembexa was studied in animals infected with viruses that are closely related to the variola virus.

Please define EIA in your first use (line 241 instead of in line 263).

Revise the sentence “Sapovirus, a nonenveloped, single-stranded RNA virus, has become (I would suggest: has been identified as) an important cause of gastroenteritis in immunocompromised patients after the introduction of reverse transcription-PCR”.

In line 280 “it may (be?) used for ...”

Line 295 “Picobirnaviruses were detected in up to 40% of stool samples by RT‐PCR assays ...” Please define: Where? When?

Revise line 295 “...was associated with to (sic)  high...”

Finally, I would suggest the inclusion and discussion of some references to enrich the manuscript:

Bueno et al. J Med Virol. 2021 Aug;93(8):4773-4779. doi: 10.1002/jmv.26892

Kawaguchi et al. Intern Med. 2020 Oct 15;59(20):2565-2569. doi: 10.2169/internalmedicine.4822-20.

King et al. Transpl Infect Dis. 2019 Jun;21(3):e13092. doi: 10.1111/tid.13092.

Schultze-Florey et al. Transpl Infect Dis. 2020 Feb;22(1):e13201. doi: 10.1111/tid.13201

Reviewer 2 Report

Abbas N et al focus their review article on viral agents linked to cases of gastroenteritis in immunocompromised solid organ transplant (SOT) recipients. They discuss viruses which have been well-described in connection with gastrointestinal disease in SOT patients, as well as “emergent” viruses which have been brought to light because of more sensitive detection techniques. The novelty of this paper is precisely due to inclusion of the “emergent” viruses.

Major considerations:

  1. Presentation: The abstract is a well-written summary of the paper. However, beyond this point the quality of the writing is inconsistent and major editing is required. There are stylistic errors in terms of manuscript layout, as well as misspellings, punctuation issues, word omissions, and other inconsistencies. With this many errors, it is difficult to focus on content.
  2. Virus names should respect convention. According to the International Committee on the Taxonomy of Viruses, virus names should not be italicized when used generically, nor should they be capitalized unless they contain a person’s name or a place. The authors followed convention in the Abstract but not in the Introduction nor in lines 109-133 where Norovirus should be written with a lower-case letter. However, when there is specific reference (in a taxonomic sense) to a virus family, subfamily, genus or species, then the virus name should be italicized and capitalized. The entire manuscript should be revised and corrections made where appropriate.

- Further, on line 35, the word “beta-Herpesviridae” does not refer to a subfamily. The subfamily should be written as Betaherpesvirinae.

- Line 100: “Calciviridae” should be changed to “Caliciviridae”.  

  1. Acronyms must be defined at first use of the word. For example, on line 79 the acronym “GCV” is used in place of ganciclovir. GCV should appear on line 68 after the first use of ganciclovir, after which only “GCV” should be used.
  2. In terms of content, I would suggest that there be more consistency in the information pertaining to the general characteristics of the viruses included in the review. It makes good sense to include the type of nucleic acid, whether it is single- or double-stranded and, in cases where the nucleic acid is ssRNA, then there should be mention of whether it is a positive or negative strand. The presence, or not, of a viral envelope is also an important characteristic of the virus. The virus family should also be mentioned consistently.
  3. The section on adenoviruses might benefit by adding information of the specific genus and species known to be pathogenic in immunosuppressed SOT patients.
  4. While the review focusses on major viral pathogens causing enteritis in SOT patients, I believe that it might be of interest to add the Epstein-Barr virus seeing that there are a few reports of EBV enteritis in intestinal transplant patients (for example see Pascher A et al, Transpl Proc 2004, 36:381. Lau AH et al, Pediatr Transpl 2010, 14:544).

Author Response

Thank you for your time and your very helpful suggestions. Please see responses below in red.

Abbas N et al focus their review article on viral agents linked to cases of gastroenteritis in immunocompromised solid organ transplant (SOT) recipients. They discuss viruses which have been well-described in connection with gastrointestinal disease in SOT patients, as well as “emergent” viruses which have been brought to light because of more sensitive detection techniques. The novelty of this paper is precisely due to inclusion of the “emergent” viruses.

Major considerations:

  1. Presentation: The abstract is a well-written summary of the paper. However, beyond this point the quality of the writing is inconsistent and major editing is required. There are stylistic errors in terms of manuscript layout, as well as misspellings, punctuation issues, word omissions, and other inconsistencies. With this many errors, it is difficult to focus on content.

Grammatic, stylistic inconsistencies and misspellings were corrected. Apologies for any oversight.

  1. Virus names should respect convention. According to the International Committee on the Taxonomy of Viruses, virus names should not be italicized when used generically, nor should they be capitalized unless they contain a person’s name or a place. The authors followed convention in the Abstract but not in the Introduction nor in lines 109-133 where Norovirus should be written with a lower-case letter. However, when there is specific reference (in a taxonomic sense) to a virus family, subfamily, genus or species, then the virus name should be italicized and capitalized. The entire manuscript should be revised and corrections made where appropriate.

The virus names have been corrected to follow convention.

- Further, on line 35, the word “beta-Herpesviridae” does not refer to a subfamily. The subfamily should be written as Betaherpesvirinae.

Corrected.

- Line 100: “Calciviridae” should be changed to “Caliciviridae”.

Corrected.

  1. Acronyms must be defined at first use of the word. For example, on line 79 the acronym “GCV” is used in place of ganciclovir. GCV should appear on line 68 after the first use of ganciclovir, after which only “GCV” should be used.

Corrected.

  1. In terms of content, I would suggest that there be more consistency in the information pertaining to the general characteristics of the viruses included in the review. It makes good sense to include the type of nucleic acid, whether it is single- or double-stranded and, in cases where the nucleic acid is ssRNA, then there should be mention of whether it is a positive or negative strand. The presence, or not, of a viral envelope is also an important characteristic of the virus. The virus family should also be mentioned consistently.

Information about each virus type, single or double stranded and enveloped or non enveloped was included as was virus family.

  1. The section on adenoviruses might benefit by adding information of the specific genus and species known to be pathogenic in immunosuppressed SOT patients.

Added.

  1. While the review focusses on major viral pathogens causing enteritis in SOT patients, I believe that it might be of interest to add the Epstein-Barr virus seeing that there are a few reports of EBV enteritis in intestinal transplant patients (for example see Pascher A et al, Transpl Proc 2004, 36:381. Lau AH et al, Pediatr Transpl 2010, 14:544)

A short section on EBV was included with these references and another one as well.

Round 2

Reviewer 2 Report

The revised manuscript is improved, however there are still numerous errors that require attention. Below is a list of the corrections to be made. I may not have caught everything, reason for which I again recommend a proof-reader before sending in the final copy.

  • Line 37: The word Betaherpesviridae (spelled with the “viridae” suffix) designates the virus family. To designate the subfamily, the correct word is Betaherpesvirinae (spelled with the “virinae” suffix).
  • Lines 45-46: The sentence should read as, “Recipients of lung, composite tissue, and small bowel are at particularly high risk for CMV disease due to….”.
  • Line 60: Change to, “Assessment of the level of immune response and risk for…”.
  • Line 68: Place period after reference [4].
  • Line 81: Place period after reference [19].
  • Lines 108-109: Change “symptoms” to “symptom”.
  • Line 127: Place period after reference [27].
  • Line 132: Sentence should be changed to, “Therapy with oral and intravenous immunoglobulin has generated…”.
  • Line 140: Add period after reference [30].
  • Line 141: The sentence should read as, “….but further studies are needed to establish….”.
  • Line 156 and line 158: The abbreviations ACR and GALT are unnecessary because the words they represent are only used once.
  • Line 166: Change “vaccine” to “vaccines”.
  • Line 184: Add a period after reference [38].
  • Line 195: Change “rage” to “range”.
  • Line 200: The sentence should read as, “Patients with adenoviral infection may clear it on their own, but those…”.
  • Line 203: The sentence should read as, “….and no agents have been approved for treatment of ….”.
  • Line 205: Cidofovir should be abbreviated to CDV seeing that it was already defined in the prior text.
  • Line 218: Use CDV abbreviation.
  • Line 229: Use CDV abbreviation.
  • Lines 238-239: Gammaherpesvirinae is a subfamily.
  • Line 240: Write as, “…EBV establishes latency in its host and….”.
  • Line 250: I suggest designating “EMERGENT VIRAL INFECTIONS CAUSING GASTROENTERITIS” as a new heading (3), after the Introduction (1) and Viral Enteritis (2), with the Conclusion becoming heading 4. As such, it will be in bold with only the first letter of each word in capital letters and no underlining. Each of the featured viruses can then follow the same layout as the viruses featured in section 2.
  • Line 253: Change “astroviruses” to “astrovirus”.
  • Line 260: As explained above, “Human Astroviruses” should be a sub-heading under section 3. Follow same form as viruses included in section 2 subheadings. The paragraph under this subheading can begin as follows, “Human astroviruses (HastV) belong to the family Astroviridae, and are positive-sense, single-stranded RNA viruses divided into classic….”.
  • Line 272: change to, “Several commercial enzyme immunoassays (EIAs)…”.
  • Line 278: Same comment as for line 260. Subheading should be “Human Bocaviruses” (unbolded, underlined). The paragraph under this subheading can begin as, “Human bocaviruses (HBoV) are non-enveloped, single-stranded DNA viruses belonging to the family Parvoviridae. HBoV can cause respiratory infections and gastroenteritis in immunocompromised patients [51]. There are four HBoV genotypes…..”.
  • Line 284: Write “establish” instead of “established”.
  • Line 291: Write “a course later complicated….”.
  • Line 294: Write “diagnosis” instead of “diagnostic”.
  • Line 298: As previously stated, “Human Cosaviruses” should be a subheading (unbolded and underlined). Then follow with, “Human cosaviruses (HCoV) are members of the family Picornaviridae. They are positive-sense single-stranded RNA viruses classified into six species….”.
  • Line 308: Subheading is “Sapoviruses” (unbolded and underlined). Follow with “Sapoviruses are members of the Caliciviridae. They are non-enveloped single-stranded positive-sense RNA viruses which have been identified…”.
  • Line 315: Subheading is “Toroviruses” (unbolded and underlined). Follow with, “Toroviruses are members of the Tobaniviridae family. They are enveloped positive-stranded RNA viruses, which are established agents of gastroenteritis in animals. Toroviruses are now a recognized cause of gastroenteritis in pediatric patients….”.
  • Line 323-324: Sentence should read, “…the diagnosis is difficult to make since most centers…”.
  • Line 326: Subheading is “Picobirnaviruses” (unbolded and underlined). Follow with, “Picobirnaviruses are members of the Picobirnaviridae. These viruses are small non-enveloped bi-segmented double-stranded RNA viruses which can cause diarrhea…”.
  • Line 332: Add a period after reference [65].
  • Line 333: Subheading is “Severe Acute Respiratory Syndrome Coronavirus 2”. Follow with, “The severe acute respiratory syndrome coronavirus 2 (SARS-CoV-2) is a member of the Coronaviridae. SARS-CoV-2 is an enveloped positive-sense single-stranded RNA virus which may play a role in …”
  • Line 336: Use abbreviation “SOT” instead of “solid organ transplant”.

Author Response

Thank you very much for your time and comments on the manuscript. All of the suggested changes have been made and the manuscript has been proofread. Please see below comments in red indicating completion of revision.

The revised manuscript is improved, however there are still numerous errors that require attention. Below is a list of the corrections to be made. I may not have caught everything, reason for which I again recommend a proof-reader before sending in the final copy.

  • Line 37: The word Betaherpesviridae (spelled with the “viridae” suffix) designates the virus family. To designate the subfamily, the correct word is Betaherpesvirinae (spelled with the “virinae” suffix). Done.
  • Lines 45-46: The sentence should read as, “Recipients of lung, composite tissue, and small bowel are at particularly high risk for CMV disease due to….”.  Done
  • Line 60: Change to, “Assessment of the level of immune response and risk for…”.  Done
  • Line 68: Place period after reference [4].  Done
  • Line 81: Place period after reference [19]. Done
  • Lines 108-109: Change “symptoms” to “symptom”.  Done
  • Line 127: Place period after reference [27].  Done
  • Line 132: Sentence should be changed to, “Therapy with oral and intravenous immunoglobulin has generated…”.  Done
  • Line 140: Add period after reference [30].  Done 
  • Line 141: The sentence should read as, “….but further studies are needed to establish….”.  Done
  • Line 156 and line 158: The abbreviations ACR and GALT are unnecessary because the words they represent are only used once.  Done
  • Line 166: Change “vaccine” to “vaccines”.  Done
  • Line 184: Add a period after reference [38]. Done
  • Line 195: Change “rage” to “range”. Done
  • Line 200: The sentence should read as, “Patients with adenoviral infection may clear it on their own, but those…”. Done
  • Line 203: The sentence should read as, “….and no agents have been approved for treatment of ….”. Done
  • Line 205: Cidofovir should be abbreviated to CDV seeing that it was already defined in the prior text.  Done
  • Line 218: Use CDV abbreviation.  Done
  • Line 229: Use CDV abbreviation.  Done
  • Lines 238-239: Gammaherpesvirinae is a subfamily.  Done
  • Line 240: Write as, “…EBV establishes latency in its host and….”.  Done
  • Line 250: I suggest designating “EMERGENT VIRAL INFECTIONS CAUSING GASTROENTERITIS” as a new heading (3), after the Introduction (1) and Viral Enteritis (2), with the Conclusion becoming heading 4. As such, it will be in bold with only the first letter of each word in capital letters and no underlining. Each of the featured viruses can then follow the same layout as the viruses featured in section 2. Done
  • Line 253: Change “astroviruses” to “astrovirus”. Done
  • Line 260: As explained above, “Human Astroviruses” should be a sub-heading under section 3. Follow same form as viruses included in section 2 subheadings. The paragraph under this subheading can begin as follows, “Human astroviruses (HastV) belong to the family Astroviridae, and are positive-sense, single-stranded RNA viruses divided into classic….”.  Done
  • Line 272: change to, “Several commercial enzyme immunoassays (EIAs)…”. Done
  • Line 278: Same comment as for line 260. Subheading should be “Human Bocaviruses” (unbolded, underlined). The paragraph under this subheading can begin as, “Human bocaviruses (HBoV) are non-enveloped, single-stranded DNA viruses belonging to the family Parvoviridae. HBoV can cause respiratory infections and gastroenteritis in immunocompromised patients [51]. There are four HBoV genotypes…..”. Done
  • Line 284: Write “establish” instead of “established”. Done
  • Line 291: Write “a course later complicated….”. Done
  • Line 294: Write “diagnosis” instead of “diagnostic”. Done
  • Line 298: As previously stated, “Human Cosaviruses” should be a subheading (unbolded and underlined). Then follow with, “Human cosaviruses (HCoV) are members of the family Picornaviridae. They are positive-sense single-stranded RNA viruses classified into six species….”. Done
  • Line 308: Subheading is “Sapoviruses” (unbolded and underlined). Follow with “Sapoviruses are members of the Caliciviridae. They are non-enveloped single-stranded positive-sense RNA viruses which have been identified…”.  Done
  • Line 315: Subheading is “Toroviruses” (unbolded and underlined). Follow with, “Toroviruses are members of the Tobaniviridae family. They are enveloped positive-stranded RNA viruses, which are established agents of gastroenteritis in animals. Toroviruses are now a recognized cause of gastroenteritis in pediatric patients….”.  Done
  • Line 323-324: Sentence should read, “…the diagnosis is difficult to make since most centers…”.  Done
  • Line 326: Subheading is “Picobirnaviruses” (unbolded and underlined). Follow with, “Picobirnaviruses are members of the Picobirnaviridae. These viruses are small non-enveloped bi-segmented double-stranded RNA viruses which can cause diarrhea…”.  Done
  • Line 332: Add a period after reference [65].  Done
  • Line 333: Subheading is “Severe Acute Respiratory Syndrome Coronavirus 2”. Follow with, “The severe acute respiratory syndrome coronavirus 2 (SARS-CoV-2) is a member of the Coronaviridae. SARS-CoV-2 is an enveloped positive-sense single-stranded RNA virus which may play a role in …” Done
  • Line 336: Use abbreviation “SOT” instead of “solid organ transplant”.  Done